# Preconditioning to Water Deficit Helps *Aloe vera* to Overcome Long-Term Drought during the Driest Season of Atacama Desert

**DOI:** 10.3390/plants11111523

**Published:** 2022-06-06

**Authors:** José P. Delatorre-Castillo, José Delatorre-Herrera, Kung Sang Lay, Jorge Arenas-Charlín, Isabel Sepúlveda-Soto, Liliana Cardemil, Enrique Ostria-Gallardo

**Affiliations:** 1Faculty of Renewable Natural Resources, Desert Agriculture Area, Universidad Arturo Prat, Iquique 1100000, Chile; josdelat@unap.cl (J.P.D.-C.); jodelato@unap.cl (J.D.-H.); kungsanglay@gmail.com (K.S.L.); jarenas@unap.cl (J.A.-C.); issepulv@unap.cl (I.S.-S.); 2Plant Molecular Biology Center, Department of Biology, Faculty of Sciences, Universidad de Chile, Santiago de Chile 7800003, Chile; lcardemi@uchile.cl; 3Laboratory of Plant Physiology, Center of Advanced Studies in Arid Zones (CEAZA), La Serena 1710088, Chile

**Keywords:** succulence, CAM, water deprivation, oxygen evolution, hyper-arid

## Abstract

Throughout evolution, plants have developed different strategies of responses and adaptations that allow them to survive in different conditions of abiotic stress. *Aloe vera* (L.) Burm.f. is a succulent CAM plant that can grow in warm, semi-arid, and arid regions. Here, we tested the effects of preconditioning treatments of water availability (100, 50, and 25% of soil field capacity, FC) on the response of *A. vera* to prolonged drought growing in the hyper-arid core of the Atacama Desert. We studied leaf biomass, biochemical traits, and photosynthetic traits to assess, at different intervals of time, the effects of the preconditioning treatments on the response of *A. vera* to seven months of water deprivation. As expected, prolonged drought has deleterious effects on plant growth (a decrease of 55–65% in leaf thickness) and photosynthesis (a decrease of 54–62% in E_max_). There were differences in the morphophysiological responses to drought depending on the preconditioning treatment, the 50% FC pretreatment being the threshold to better withstand prolonged drought. A diurnal increase in the concentration of malic acid (20–30 mg mg^−1^) in the points where the dark respiration increased was observed, from which it can be inferred that *A. vera* switches its C3-CAM metabolism to a CAM idling mode. Strikingly, all *A. vera* plants stayed alive after seven months without irrigation. Possible mechanisms under an environmental context are discussed. Overall, because of a combination of morphophysiological traits, *A. vera* has the remarkable capacity to survive under severe and long-term drought, and further holistic research on this plant may serve to produce biotechnological solutions for crop production under the current scenario of climatic emergency.

## 1. Introduction

Succulence is the main adaptive syndrome for water storage in plant tissues [1]. Succulence allows plants to uncouple temporarily from the external water supply [2]. Succulent plants occur globally, although they are mostly frequent in hot and dry ecosystems with marked seasonality [3,4]. Water-storing tissues can occur in any part of the plant, namely root, stem, or leaves. Particularly for photosynthetic succulent leaves, there are two types, according to its anatomical innovation: (a) all-cell succulence; (b) storage succulence (further details in [5]. *Aloe vera* (L.) Burm.f. (Asphodelaceae) is a storage succulent type, with a central core of hydrenchyma surrounded by a peripheral chlorenchyma in which photosynthesis occurs. Along with succulence, *A. vera* utilizes the crassulacean acid metabolism (CAM) for carbon assimilation [6]. Because of its remarkable medicinal benefits, *A. vera* has been globally cultivated for centuries. The adaptive features associated with succulence and carbon gain, plus its anti-inflammatory and bioactive compounds make *A. vera* one of the most attractive and profitable alternative crops for marginal dry lands, particularly in the context of the current global change [7,8,9].

The set of morphophysiological traits of *A. vera* confers adaptation to arid environments [10]. During the last few years, several studies have contributed to increasing our understanding of the adaptive responses of *A. vera* to drought stress [11,12]. The interaction of genes and gene networks regulates the flexibility of the phenotypic response of *Aloe vera* to drought. The recent sequencing and functional analysis of *Aloe vera*’s transcriptome and genome revealed specific genes with signals of adaptive evolution to drought and molecular pathways associated with drought stress tolerance [9,13]. Biochemical approaches have shown that under different levels of water stress, the acclimation of CAM processes, chloroplast ultrastructure, and photochemical activity is related to higher biosynthesis of phenolic compounds [14] and the antioxidant status [15]. Moreover, having flexible structural traits helps *Aloe vera* cope with dehydration events. In this sense, [16] studied the dynamic of cell-wall folding during seasonal drought stress in Aloe species. They found that the folding of the hydrenchymatic cells is a highly regulated process, in which the function and content of mannans and pectins are particularly key during the seasonal drought and drought recovery. The authors also suggest a role of mannans and pectins in the hydrenchyma-chlorenchyma water movement, being apparently a mechanism used by plants to maintain photosynthesis under stress conditions [17]. Besides the advantage conferred by its morphophysiological traits, the intrinsic adaptive responses of *A. vera* ca be expanded with the application of biofertilizers, as demonstrated by [18]. The authors showed that when applying a bioproduct of mycorrhizal fungi and phosphate-solubilizing bacteria to drought-stressed *A. vera* plants, the fresh weight of the leaves, total chlorophyll content, and photosynthetic yield were similar to non-stressed plants. Strong osmotic regulation allows *A. vera* to endure very dry climatic conditions. In the hyper-arid Atacama Desert, [11] kept plants of *A. vera* under different irrigation regimes in respect to the soil field capacity (100, 50, and 25% FC). The plants kept at 50% FC showed a significant increase of proline biosynthesis, total soluble sugars, and fructans. The authors point out that, overall, the increase of osmolytes causes water retention, buffering the water loss from leaves, and, under very xeric environmental conditions, moderate drought stress enhances the WUE of *A. vera*.

Despite having highly adapted traits to cope with drought, the extent of the adaptive responses of *A. vera* depends on the severity and time of exposure to drought. For example, [14] evaluated several growth and physiological traits of *A. vera* at different time intervals for 180 days, in response to moderate and severe drought stress (80 (control), 40 and 20% of the field capacity). Significant differences among biomass and physiological traits were observed from day 60 of drought stress. In particular, the values of the index related to biomass, cell protection, and maximum quantum yield (Fv/Fm) and performance index (PIabs) for photosynthetic yield of plants at 40% FC were similar or even higher than control plants by the end of the experiment. However, the opposite was observed for the severe drought treatment (20% field capacity), showing the lowest values for biomass yield, Fv/Fm, and PIabs by the end of the experiment.

The hypothesis is that *A. vera*, depending on the water content accumulated in its foliar tissues, can maintain vital processes, including photosynthesis. This will allow, in conditions of prolonged water deficit, the plants to keep the photosynthesis process operational, even if they present lower photochemical, morphological, and/or biochemical efficiency.

Here, *A. vera* plants were preconditioned to different levels of field capacity of the soil (100, 50, and 25% FC) for 3 months, and then were subjected to a prolonged drought during the driest season of the Atacama Desert (7 months) (Figure 1). We studied leaf biomass, biochemical traits, and photosynthetic traits to assess, at different intervals of time, the effects of the preconditioning treatments on the response of *A. vera* to prolonged drought. Overall, despite the variability of responses associated with the preconditioning treatments, all *A. vera* plants remained metabolically active during 7 months without irrigation in one of the driest places on Earth. Possible mechanisms and the environmental context are discussed.

## 2. Results

### 2.1. Leaf Biomass and Biochemical Indicators of Stress

#### 2.1.1. Chlorenchyma and Hydrenchyma Thickness

In general, all plants showed a decrease in leaf thickness by the end of the experiment (Figure 2A,B). However, the prolonged water stress (days of water deprivation, DWD) has distinct effects on the change in thickness of the adaxial and abaxial portions of chlorenchyma. The adaxial portion showed a sustained and significant decrease of thickness along the whole DWD period. Overall reduction was of 22.5% from day 88 up to 222 DWD. In contrast, thickness of the abaxial chlorenchyma showed a significant decrease only at the end of the experiment (222 DWD), and the reduction from day 88 up to day 222 was of 8.9%. On the other hand, the hydrenchyma portion of the leaves (Figure 2C) showed higher thickness at 88 DWD, and then decreased significantly from 118 DWD up to the end of the experiment at 222 DWD. Overall reduction of the hydrenchyma portion was of 35.9% from day 88 up to day 222, which is also the largest reduction of thickness among leaf sections.

Preconditioning treatments showed differences in thickness depending on the leaf portion and the period of DWD. For both adaxial and abaxial portions of chlorenchyma, plants preconditioned at 25% FC were significantly thinner (see Appendix A) than those preconditioned at 50 and 100% FC, but this difference was significant only at 222 DWD. The effects of preconditioning on the hydrenchyma thickness showed significant differences earlier than in chlorenchyma, at 118 DWD. First, preconditioned plants from 25 and 50% FC were significantly lower than 100% FC at 118 DWD. At 222 DWD, preconditioned plants from 25% FC showed the thinnest value, whilst those from 50 and 100% FC showed similar thickness values (Figure 2A–C).

#### 2.1.2. Chlorenchyma and Hydrenchyma Biomass

The chlorenchyma biomass responded differently regarding the portion of the leaf, i.e., adaxial or abaxial. Overall decrease of the adaxial weight was of 12.5% from day 88 up to day 222. The adaxial weight decreased significantly at 222 DWD, and no significant effects of the preconditioning treatments were observed on the biomass changes along the whole experiment. On the other hand, the abaxial weight did not show differences regarding the duration of drought stress, decreasing only 1.8% from day 88 up to day 222. However, plants preconditioned at 25% FC showed a significant decrease of biomass compared to those from 50 and 100% FC at the end of 222 DWD. In contrast, the hydrenchyma biomass showed a similar trend and a significant change to that of thickness. Overall reduction from day 88 up to day 222 was of 44.9%. Plants showed a significant decrease of hydrenchyma weight at 118 DWD, but then the weight remained similar up to 222 DWD. There was evidence for significant effects of preconditioning at the end of the experiment, with the 25% FC the lowest at 222 DWD, whilst no significant differences were found between plants from 50 and 100% FC treatments (Figure 2D–F).

#### 2.1.3. Proline and MDA

Proline quantification did not show evidence for significant differences among preconditioning treatments and during the course of prolonged drought stress. Proline values were below 0.5 mg g^−1^ dry weight and remained similar through the experiment (Figure 3A).

MDA quantification was high and showed significant differences between 118 and 222 DWD, the latest with the highest values. From day 118 up to day 222, there was an overall increase of 24.1%, averaging 530.99 nmol g^−1^ of MDA. There was no evidence for significant differences between preconditioning treatments within any specific time interval of DWD. However, significant differences appeared when comparing each preconditioning treatment through the course of prolonged drought. Specifically, preconditioned plants from 50 and 100% FC showed significant increase of MDA from 118 to 222 DWD (Figure 3B).

### 2.2. Photosynthetic Characteristics of Aloe vera under Prolonged Drought Stress

#### 2.2.1. E/PFD Curves

The response curves to light are presented in Figure 4; it can be seen how the rates of photosynthesis (E) decline as a function of time and the initial volume of the leaves; this is a product of the irrigation treatment that the plants received before subjecting them to the water deficit. 

Rate of photosynthetic evolution (Emax) at 222 days (Figure 4C,F,J) declined depending on the availability of water accumulated in their tissues, and that is related to the volume of the tissues. The longer the duration of water deficit, the treatment that received only irrigation at 25% CdC reduced its maximum assimilation rate to 50%.

#### 2.2.2. Light Compensation and Saturation Points

Light compensation and saturation points changed during the experiment (Figure 5A,B). Light compensation point (LCP) showed significant increase of 139.7% from day 118 up to day 222. Preconditioning treatments also showed evidence of significant effects. At the end of the experiment (222 DWD), the LCP of plants from 25% FC was higher than those from 50 and 100% FC. On the other hand, light saturation point showed a sustained increase during the whole period of prolonged drought stress, averaging 424.39 µmol photons m s^−1^ at 222 DWD. However, there were no differences associated with the preconditioning treatments along the experiment. Both parameters had a strong relationship. A linear regression analysis showed an R^2^ of 0.92 (Appendix B). Depicting this relationship by the time of water stress by separate means, it is possible to observe that the regression coefficient is higher at the end of the experiment (R^2^ 0.94) (Appendix B).

#### 2.2.3. Apparent Quantum Yield (AQY), Photosynthetic Capacity (Emax), and Respiration (Rd)

In general, parameters associated with photosynthetic oxygen evolution tended to decrease under prolonged water stress (Figure 6A–C). There is evidence for a significant decrease in the apparent quantum yield at 222 DWD compared with 88 DWD. In fact, the apparent quantum yield showed an overall decrease of 46% from day 88 up to day 222. Differences in AQY related to the preconditioning treatment were observed only at 88 DWD (Figure 6A). Despite a tendency to decrease during the experiment and an overall reduction in 41.8%, the photosynthetic oxygen evolution (Emax) did not show evidence for significant change during the prolonged water stress or among preconditioning treatments (Figure 6B). Dark respiration (Rd) showed a significant decrease of 25.9% from 88 DWD to 118 DWD, but without further significant decrease up to the end of the experiment at 222 DWD. Differences in Rd related to preconditioning treatments were observed only at 88 DWD between 25 and 100% FC plants (Figure 6C). Emax and Rd had a positive and negative relationship with AQY, respectively (R^2^ 0.68 and R^2^ 0.58) (Appendix C).

#### 2.2.4. Malic Acid Evolution and Accumulation during Prolonged Drought Stress

Malic acid (MAc) decreased significantly by the end of the experiment, with an overall decrease of 26.7% from day 88 up to day 222. Moreover, the effects of the preconditioning treatments showed significant difference only at the end of the experiment, where the 25% FC preconditioned plants showed the lowest accumulation of MAc.

There were differences for daily MAc evolution through the course of the experiment, depending on the time interval and preconditioning treatment (Figure 7A,B). At 88 DWD, the MAc evolution showed an increase at 2 a.m. with peaks near to midday. At 118 DWD, a first increase of MAc occurred from 22 h, followed by a slight decrease at 6 a.m., and then an increase with peaks at midday (14 p.m.). Finally, at 222 DWD, preconditioned plants from 50 and 100% FC showed an increase of MAc near to 2 a.m., with peaks at 10 a.m. On the other hand, plants from 25% FC preconditioning treatment showed an increase from 10 a.m. with a peak at 14 p.m. Moreover, the amount of MAc during the rest of the day was lower compared to plants preconditioned at 50 and 100% FC.

## 3. Discussion

As expected, leaf biomass and photosynthesis decreased with the progress of drought stress. However, after 7 months without irrigation, in one of the most xeric places on Earth, all *A. vera* plants stayed alive. Moreover, we found evidence that the preconditioning treatments had different effects on the degree of response to prolonged water deprivation.

The storage-succulent-type leaf of *Aloe vera* is undoubtedly a major adaptive trait for living under water stress conditions. The large central hydrenchyma allows the storage of a high volume of water and metabolites, whereas the peripherical chlorenchyma has mainly a photosynthetic function [1]. In our experiment, the preconditioning treatments (100, 50, and 25% FC) presuppose an acclimation process for water storage and for a physiological adjustment in the function of water availability. After the first three months of preconditioning, all plants had similar growth parameters despite the differences in watering, highlighting the conservative but high water use efficiency strategies of *A. vera* under the variation of water availability [10,11]. Hence, it may be assumed that the more water availability the plant has, the more water would be stored in the hydrenchyma, and the better the drought buffering would be [1]. *A. vera* belongs to the group of succulents identified as drought-avoidance plants, which activate mechanisms for water translocation from the hydrenchyma to the chlorenchyma during seasonal drought to sustain carbon metabolism [19]. Our results suggest that water translocation from the hydrenchyma to the chlorenchyma occurs, but is not equitable, prioritizing the abaxial over the adaxial chlorenchyma. As the adaxial surface receives more solar radiation than the abaxial surface, there would be more evaporative and photo-oxidative pressure for the former. [10] found significant differences for the lengths and widths of occlusive cells between both surfaces, being longer and narrowed in the adaxial surface. This allows higher stomata resistance to the adaxial portion against the strong evaporative demand and solar radiation. In accordance with the above mentioned, our results suggest a compensation mechanism for the loss of biomass and thickness of the adaxial surface. The maintenance of thickness and biomass reported here for the abaxial surface plus a lower stomatal resistance [10] would be key to coping with prolonged drought in xeric environments. Since mannans and pectins are involved in the hydrenchyma–chlorenchyma water movement [17], how these sugars would be involved in differential movement to the abaxial face of the leaf needs to be solved. At the end of the prolonged drought period, plants preconditioned at 50% FC showed similar values for adaxial and abaxial chlorenchyma thickness and weight to those preconditioned at 100% FC. This is typical for succulent CAM plants in which an excess of water does not produce better growth or physiological yields [20]. Hence, it is plausible that drought-avoidance and drought-tolerance strategies can coexist under prolonged drought [21,22]. In our study, the results suggest that the drought-avoidance strategy predominates up to day 88 of drought, and then would coexist with a drought-tolerance strategy up to the end of the prolonged drought treatment, prioritizing water translocation to the abaxial portion.

As mentioned above, plants watered with 100% of field capacity of the soil prior to the experimental prolonged drought do not perform better than plants preconditioned at 50% FC. Indeed, the MDA levels, utilized as a stress indicator, were slightly higher in the 100% FC preconditioned plants. The latest seems to be a recurrent response in *A. vera* and other succulent CAM species, in which the extremes of watering deviate plant performance from the optimal [11,14,20]. Moreover, it is worth noting that *A. vera* possesses several other compounds (phenolics, flavonoids, and secondary metabolites) with high radical scavenging activity [23], that would be involved in membrane protection under the period of water deprivation. Moreover, change in pigmentation during the course of water deprivation (Appendix A) reflects the increase of “sunscreen” pigments to protect leaf ultrastructure from photo-oxidative damage. Although MDA levels were high, especially from day 118, the overall antioxidant activity of *A. vera* would be a major mechanism used by this plant to withstand and tolerate prolonged drought periods. Additionally, despite there being no evidence for significant differences, proline tended to increase in plants preconditioned at 50 and 25% FC during the progress of drought whilst the opposite occurred in those preconditioned at 100% FC. Proline is known as a first-line response against drought stress. Hence, the induction of mechanisms of enzymatic and non-enzymatic defenses [24,25] would be involved in the long-term drought in *A. vera*, being a plausible explanation for the similar levels of MDA found between days 88 and 222 of prolonged drought, and the decrease observed at day 118.

Despite its decrease, all *A. vera* showed both photosynthetic and respiration activity up to the end of the prolonged drought. This is remarkable, considering that the experiment was carried during the driest season at the hyper-arid core of the Atacama Desert. *A. vera* is a succulent CAM plant. For CAM metabolism, the enzyme phosphoenol-pyruvate carboxylase (PEPC) is key. The activation/deactivation of PEPC is regulated by a circadian control of phosphorylation and dephosphorylation. The active site of PEPC is specific to CO_2_ and has a light-independent activity; thus, it can fix CO_2_ in the dark. The convergence of CAM and succulence is a functional advantage for saving water. However, the relationship between succulence and the biochemical rhythms of CAM still needs to be better understood, especially under different climatic pressures. The climatic conditions of the Atacama Desert impose several factors affecting water availability. In our study site, plants were exposed to harsh climatic conditions such as high solar radiation; vapor pressure deficit; thermal amplitude; and reference evapotranspiration, along with an average low relative humidity and the nearly absent precipitation (Appendix A). It is known that the circadian control and expression of CAM phases (I–IV) are modulated by the environmental conditions, especially by irradiance and water availability [26]. Phases I and II correspond to the nocturnal CO_2_ uptake by PEPC and the daytime regeneration of CO_2_ for Rubisco fixation, respectively. During the day, when nocturnally stored organic acid is already consumed, phase IV takes place. Thus, stomata may open under the light period and CO_2_ is assimilated directly by Rubisco (C3 pathway) [27]. However, under low water availability, phase IV is suppressed, especially when it is accompanied by high VPD. Moreover, under severe drought, nocturnal CO_2_ assimilation (phase I) may be delayed or even reduced [28]. Under a combination of drought, high VPD, and full-sun exposure, some CAM plants can recycle respiratory CO_2_ during the night and reassimilate it in the day, while stomata remain completely closed in both night and day [27]. This metabolic switch is known as CAM idling mode. Our results showed a delay of phase I (Figure 7B) suggesting that plants enter a CAM idling mode. CAM idling is driven by solar irradiation, and despite a net carbon gain of zero, the loss of water is greatly reduced, and in turn, plants can overcome several days, weeks, and even months of low water availability. However, the cost for the plants with the CAM idling mode is an overproduction of reactive oxygen species (ROS) and high exposure of cell components to oxidative damage [29]. As mentioned above, the radical scavenging activity [23,30] and the reddish color of the leaves observed by the end of the experiment (Appendix A) reflect mechanisms to deal with oxidative damages, and would compensate part of the cost of the CAM idling mode. In our study, prior to the preconditioning treatments, well-watered plants showed maximum oxygen evolution rates at c.a. 800 µmol photons m^−2^s^−1^. The average of the maximum irradiance during the prolonged drought period was 2475 µmol photons m^−2^s^−1^ (Appendix A), exceeding by far the initial light saturation point. In general, as consequences of a higher irradiance threshold, the circadian rhythm of CAM turns arrhythmic and the accumulation of vacuolar malic acid decreases. Our results showed that succulence and CAM helped *A. vera* to overcome 7 months of water deprivation under extreme environmental conditions. Strikingly, there was a sustained increase of light saturation point during the period of water deprivation in all *A. vera* plants, whilst light compensation point increased only at the end of the experiment. This suggests a highly regulated light acclimation process under drought stress and high irradiance. This would occur at the light harvesting system level, finely regulating the antenna size and photoprotection to PSII and PSI to deal with excess of excitation pressure and reduce photo-oxidative damage [14,25,31,32]. The sustained decrease over time in the quantum efficiency in turn generated a decrease in the Emax rates, obtaining values of 1.36 at the end of the test: 3.05 and 4.40 µmoles of O_2_ m^−2^ s^−1^. During drought stress, the plant’s light requirement is significantly reduced and excess light can damage the photosynthetic machinery, leading to photoinhibition, which is reflected in aloe with increased reddish coloration, due to an increase of rhodoxanthin [33]. The greatest damage from excess light is perceived by PSII [25]. Moreover, the deterioration of the photosynthetic metabolism can occur due to biochemical limitations [34] due to the low supply of ATP and NADPH or defects in the transport of electrons and the use of assimilation products [35].

Thus, for *A. vera*, succulence plus a CAM idling mode and strong protection/regulation of the light harvesting system would be the key process to produce long periods of water deprivation under harsh climatic conditions. Finally, it is worth noting that, although the hyper-arid core of the Atacama Desert imposes extreme climatic conditions for water availability, fog events occur regularly [36]. Known locally as “Camanchaca”, this advective fog formation is a typical climatic event in the study site, and should be an alternative, if not the regular source of water in this ecosystem. The absolute maximum relative humidity (RH) registered in the study site, reaching sometimes 90% RH, may reflect fog events (Appendix A). The use of fog as a water source by *A. vera* still needs to be solved, but it is reasonable to think that there could be another possible mechanism that allowed this plant to withstand seven months of (soil) water deprivation in one of the most arid ecosystems on Earth. In our experiment, we demonstrated that upon preconditioning plants under an intermediate field capacity of the soil, *A. vera* showed similar-or-better responses that the extreme preconditioning treatments, i.e., 100 and 25 %FC. Further research is needed to better understand the regulation of the genetic, metabolic, and physiological mechanisms of the avoidance and tolerance responses of *A. vera* to combined abiotic stress. Overall, because of a combination of morphophysiological traits, *A. vera* has a remarkable capacity to survive under severe and long-term drought, and further holistic research on this plant may serve to produce biotechnological solutions for crop production under the current scenario of climatic emergency.

## 4. Materials and Methods

### 4.1. Study Site and Preconditioning Treatment

The study site is part of the longitudinal valley of the Atacama Desert in northern Chile, known as Pampa del Tamarugal. Regional climate corresponds to a subtropical desert within the hyper-arid core, with 0.6 mm of MAP and 17.9 °C of MAT, although advective fog events occur frequently during winter and spring [36,37]. Specifically, the experiment was carried out in dependencies of the Canchones Experimental Station from the Universidad Arturo Prat (20° 26′ 37.74″ S, 69° 32′ 09.50″ W), in the community of Pozo Almonte, Tarapacá Region. Typically, poor development of horizons, saline crusts, low organic matter content, and high pH are characteristics associated with the soil properties of the study site [38].

In a 250 m^2^ plot, 3-month-old plants of *Aloe vera* (*n* = 144) were planted directly in the soil at similar depth. A drip-irrigation system kept plants well-watered daily. The emission flow was 4 l h^−1^ applied for 20 min, to reach 1.1 l day^−1^. Irrigation frequency was determined based on the crop coefficient (Kc = 0.17; Silva et al., 2010), the reference evapotranspiration (ET_0_, Appendix A), obtained from a meteorological station installed at field, and the daily evaporation. Once established, a three-month pre-experimental period of preconditioning to drought stress was carried out by modifying the irrigation frequency to 20, 10, and 5 min, achieving three levels of field capacity (FC) of the soil: 100, 50, and 25 %FC (Figure 1). Field capacity (FC) was calculated according to the model developed by [39]. Briefly, the normalized water content at field capacity (θfc) was obtained by the quotient between the drainage flux at field capacity (qfc), and the saturated hydraulic conductivity (Ks) of the soil. Soil moisture measurements were made with a ThetaProbe ML2x connected to a data logger unit model HH2 (Delta-T Devices Ltd., Cambridge, UK). Each % FC treatment had three replicates randomly distributed (details below). At the end of the preconditioning period, plant and leaf traits were recorded as a starting point before conducting the experimental prolonged water stress (Table 1).

### 4.2. Prolonged Water Stress Experiment

At field, after the preconditioning period, plants were subjected to prolonged water stress (days of water deprivation, DWD) by stopping irrigation for 222 days, from mid-spring up to mid-autumn. The experimental design was completely randomized blocks with three replications, applied from the preconditioning treatment. The experimental units consisted of a total of nine rows with sixteen plants per row (Figure 7). Sampling and measurements were made on homogenous plants, showing similar characteristics of growth and number of leaves and avoiding the border effect. Data were collected at three intervals of time, at day 88, 118, and 222 after stopping irrigation.

### 4.3. Plant Biomass and Biochemical Parameters

Thickness (cm) and biomass (g) partition of leaf tissues were obtained from foliar discs of 10 cm^2^ taken with a punch-sampler of 3.5 cm diameter. Leaf discs were sampled from the central part of the leaf on three biological replicates according to the preconditioning treatment. Changes in thickness and biomass were evaluated for the adaxial and abaxial chlorenchyma and central hydrenchyma. Leaf thickness was measured with a digital meter foot whilst biomass was measured by using a digital analytical scale with a readability of 0.005–0.1 mg.

As biochemical parameters, total proline content was measured according to [40]. Briefly, 100 µL of centrifuged supernatant extract (20 mg freeze-dried tissue plus 1 mL ethanol 70% at 4 °C) was incubated with a reaction mix of ninhydrine (1% *p*/*v*) and glacial acetic acid (20% *v*/*v*) at 95 °C for 20 min. Proline concentration was determined at 525 nm with an Epoch Microplate Spectrophotometer (Agilent, Santa Clara, CA, USA).

Lipid peroxidation in plant tissues was used as a physiological indicator of oxidative stress. Lipid peroxidation was assessed by the thiobarbituric acid method as described in [41]. This test determines the amount of malondialdehyde (MDA), a byproduct of lipid peroxidation that reacts with thiobarbituric acid. Ground frozen tissue (0.1–0.2 g) was homogenized following addition of 1 mL of TCA–TBA–HCl reagent [15% (*w*/*v*) trichloroacetic acid (TCA), 0.37% (*w*/*v*) 2-thiobarbituric acid (TBA), 0.25 M HCl, and 0.01% butylated hydroxytoluene]. After homogenization, samples were incubated at 90 °C for 30 min in a hot block, then chilled in ice and kept in darkness for 5 min, and then centrifuged at 12,000× *g* for 10 min. The resulting chromophore absorbed at 535 nm, which was evaluated using Epoch Microplate Spectrophotometer (Agilent, Santa Clara, CA, USA). Values corresponding to nonspecific absorption at 600 nm were subtracted. MDA concentration was calculated directly from the extinction coefficient ε = 156 mM^−1^ cm^−1^.

Malic acid was extracted from 100 mg freeze-dried leaf samples by 80% ethanol with 30 min. ultrasonication. Malic acid was quantified by HPLC method with chemiluminescent detection following irradiation with visible light. Extracts were chromatographed on Phenomex Luna C18 column (5 µm) with 0.1 M NaH_2_PO_4_ * H_2_O as mobile phase at column temperature of 27 °C and flow rate of 0.5 mL/min. Chromatography was monitored with a diode array detector at 210 nm.

Leaf pH was measured directly on leaf discs with a pH-meter HI99161 coupled with a FC2023 electrode (HANNA Instruments, Padua, Italy).

### 4.4. Oxygen Evolution and Photosynthetic Parameters

Changes in photosynthetic O2 evolution rate were measured with a LEAFLAB-2 gas-phase oxygen electrode (Hansatech, Pentney, King’s Lynn, UK), according to [42]. Leaf discs of 10 cm^2^ were placed on an LD2/3 electrode chamber with an S1 Oxygen Electrode Disk mounted on the base of the chamber. Light response curves (0, 10, 25, 40, 55, 70, 100, 150, 250, 450, 650, and 850 µmol m^−2^s^−1^ PPFD) were used to determine the maximum photosynthetic rate; light compensation and saturation point; quantum yield; and dark respiration (Rd). The adjustment of light intensities was generated by voltage changes through the control box of the LH36/2R LED light source. The oxygraphy temperature was maintained by connecting the chamber to a circulating water bath kept at the desired temperature. Light response curves were performed over three leaf disc replicates per preconditioning treatments during a 24 h cycle at intervals of four hours (02:00; 06:00; 10:00; 14:00; 18:00; and 22:00 h), at day 88, 118, and 222 after stopping irrigation.

### 4.5. Statistical Analysis

Data were checked for normality assumptions and variance homoscedasticity. Accordingly, mean values were compared by parametric or non-parametric analyses of variance (ANOVA). When significant differences were found, these were compared with Tukey or pairwise comparison’s test at *p* ≤ 0.05. Box plots and bar charts were used to explore and better visualize the data. All statistics and plots were made using the InfoStat software [43].

## Figures and Tables

**Figure 1 plants-11-01523-f001:**
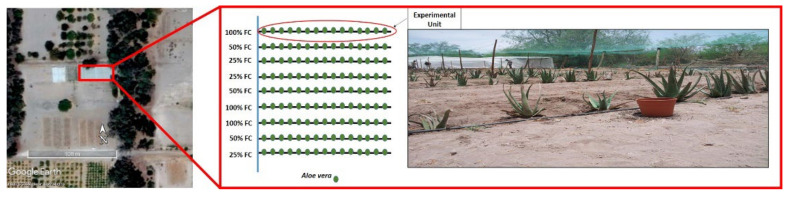
Aerial view of the study site (20° 26′ 37.74″ S, 69° 32′ 09.50″ W) with a schematic representation of the 250 m^2^ field experiment and a close-up of the experimental units analyzed in this study (details of the experimental design are in the Section 4).

**Figure 2 plants-11-01523-f002:**
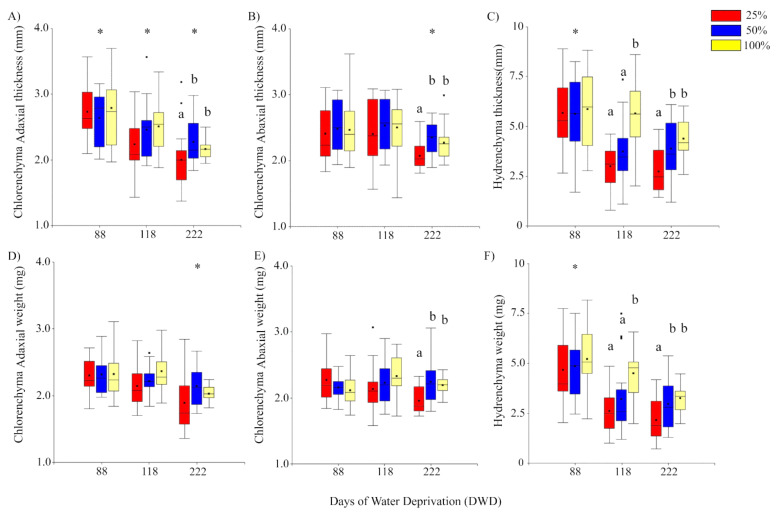
Box plots showing the changes in thickness and weight of the adaxial (**A**,**D**) and abaxial (**B**,**E**) portions of the chlorenchyma, and for the central hydrenchyma (**C**,**F**) in the function of the days with water deprivation (DWD). The central line and black dot within each box represent the average and the median. Vertical lines of the boxes indicate the upper and lower limits. Small dots outside the boxes represent extreme values. Red, blue, and yellow colors represent the values for plants preconditioned at 25, 50, and 100% of field capacity of the soil (% FC), respectively. Asterisk indicates significant differences between DWD at *p* ≤ 0.05. Different letters indicate significant differences (*p* ≤ 0.05) between preconditioning treatments within a given period of water deprivation.

**Figure 3 plants-11-01523-f003:**
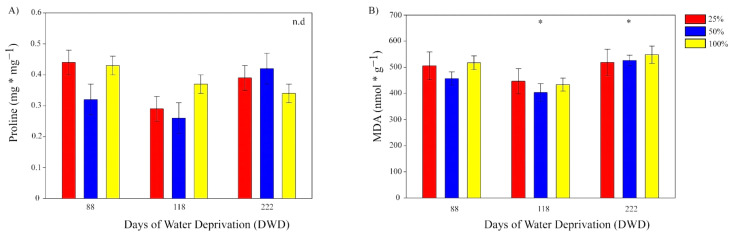
Bar chart showing the changes in (**A**). Proline content and (**B**). MDA content, in functions of the days with water deprivation (DWD). Bars correspond to the average value of each parameter, and the vertical lines indicate the standard error. Red, blue, and yellow colors represent the values for plants preconditioned at 25, 50, and 100% of field capacity of the soil (%FC), respectively. Asterisk indicates significant differences between DWD at *p* ≤ 0.05. n.d. denotes non-significant differences.

**Figure 4 plants-11-01523-f004:**
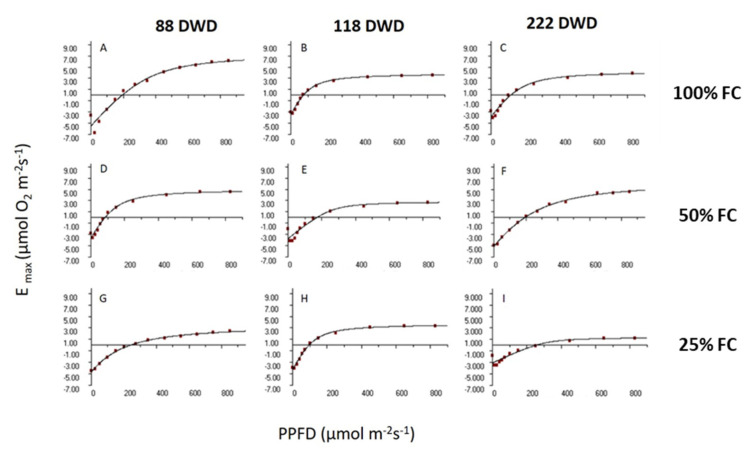
Light-responses curves for the rate of photosynthetic oxygen evolution of *A. vera* plants, previously preconditioned at 100 (**A**–**C**), 50 (**D**–**F**), and 25 (**G**–**I**) % of FC, at different intervals of time during the water deprivation experiment.

**Figure 5 plants-11-01523-f005:**
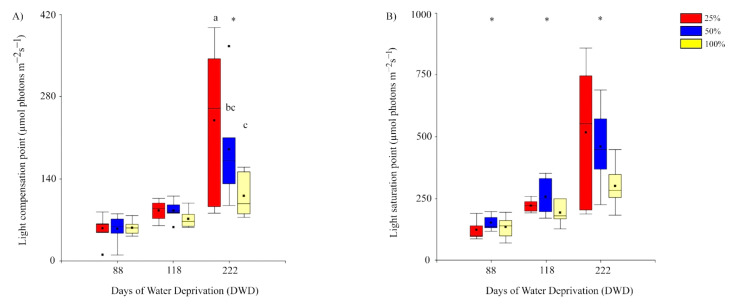
Box plots showing the changes of (**A**) light compensation and (**B**) light saturation points during the experiment of water deprivation (DWD). Lines, dots, and colors of each box correspond to what was previously indicated in Figure 2. Asterisk indicates significant differences between DWD at *p* ≤ 0.05. Different letters indicate significant differences (*p* ≤ 0.05) between preconditioning treatments within a given period of water deprivation.

**Figure 6 plants-11-01523-f006:**
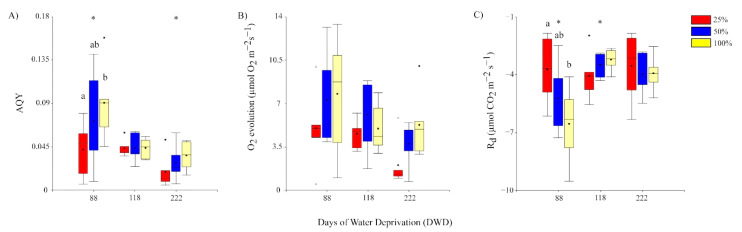
Box plots showing the changes in (**A**) apparent quantum yield (AQY); (**B**) photosynthetic capacity (O_2_ evolution); and (**C**) dark respiration (Rd), in the function of the days under water deprivation (DWD). Asterisk indicates significant differences between DWD at *p* ≤ 0.05. Different letters indicate significant differences (*p* ≤ 0.05) between preconditioning treatments within a given period of water deprivation. Lines, dots, and colors of each box correspond to what was previously indicated in Figure 2.

**Figure 7 plants-11-01523-f007:**
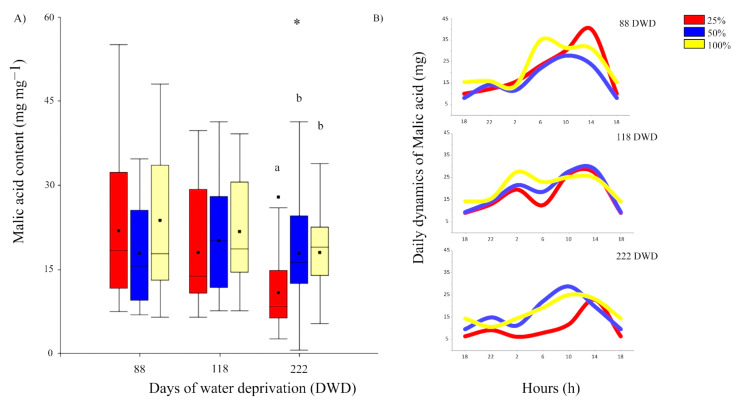
(**A**) Box plots showing the changes in malic acid content in the function of the days under water deprivation (DWD). Asterisk indicates significant differences between DWD at *p* ≤ 0.05. Different letters indicate significant differences (*p* ≤ 0.05) between preconditioning treatments within a given period of water deprivation. (**B**) Changes in daily malic acid evolution at a given interval of time under water deprivation. Lines, dots, and colors of each box correspond to what was previously indicated in Figure 2.

**Table 1 plants-11-01523-t001:** Growth parameters in Aloe plants after three months of preconditioning. Treatments showed no evidence of significant differences (*p* > 0.05).

Treatment FC	Mean Leaf N°	Leaf Thickness (cm)	Volume Leaf Disc (cm^3^)	Plant Diameter (cm)	Plant Height (cm)
25%	15.3	0.825	8.25	50.07	45.68
50%	15.4	0.794	7.94	49.00	48.10
100%	14.7	0.839	8.4	48.53	47.74

## Data Availability

Data available on request to corresponding author.

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
