# Peer review of "Preconditioning to Water Deficit Helps *Aloe vera* to Overcome Long-Term Drought during the Driest Season of Atacama Desert"

_plants, 2022, doi:10.3390/plants11111523_

Round 1
Reviewer 1 Report
Dear Authors,
I do not feel that I could agree with the main statement of the paper, that preconditioning of A. vera plants had beneficial effect on drought tolerance. After 3 types of preconditioning treatments (100, 50 and 25% FC), plants did not differ with any morfological trait measured. It suggest, that different types of analyses are neeed like physiological or moleculat to indicate traits that could have impact on drought tolerance. I did not find microscope observations that could show potential changes in chlorenchyma and hydrenhyma occured during drought period.. Moreover, it is mentioned that level of the oxidative stress under drought measured through MDA content was the highest in plants preconditioned with 100% FC. It is interesting observation that needs to be explained deeper. I would suggest to measure protein level and activity of antioxidant enzymes.
To sum up, I do not see strong conclusions of the presented studies.
Author Response
Dear Reviewer 1,
We truly appreciate all the comments made to our manuscript. We have responded to all your comments and suggestions. We feel that all the comments has helped us to clarify the information and improve the quality of our manuscript. Please see the attached file with our responses.
Kind Regards,
The corresponding author.

Reviewer 2 Report
The presented manuscript is focused on the study of the mechanism of resistance of plants with CAM metabolism to water deficit. The model plant was Aloe vera. An interesting finding is the transition between C3 and CAM metabolism. The manuscript is very interesting especially for theoretical research, with an overlap into applied research. The manuscript is written relatively carefully and meets the requirements set out in the guidelines for authors. The abstract is written carefully and summarizes the results obtained. It may be appropriate to supplement it with the values obtained. Familiarization with the issue is at the appropriate level. It is just not entirely clear to me why there is a mention of my own results at the end of this section. Please clarify. The work also focuses on the morphological features of the leaves, such as the thickness of chlorenchyme and hydranchym. I recommend not only focusing on simple measurements of physical parameters, but also creating anatomical sections. From anatomical sections, some changes related to water deficit might be more noticeable. Graph 2 is relatively small. Could it be enlarged? Graphs often have very significant standard errors. I may have overlooked it in the text, but it would be useful to focus on why this is so. The description of the results is focused on the evaluation of the condition, but without specifying the measured or relative values. The discussion is rather descriptive. It might be appropriate to relate some parameters. The methodology is adequate. Maybe I would add the dates of measurements during the day and the content of nutrients and water in the soil. Please check the citation of individual literary sources.
Author Response
Dear Reviewer 2,
We truly appreciate all the comments made to our manuscript. We have responded to all your comments and suggestions. We feel that all the comments has helped us to clarify the information and improve the quality of our manuscript. Please see the attached file with our responses.
Kind regards,
The corresponding author

Round 2
Reviewer 1 Report
Dear Authors,
thank you for your answears. I have no further comments.